# Current Status of the Diagnosis of Early-Stage Pancreatic Ductal Adenocarcinoma

**DOI:** 10.3390/diagnostics13020215

**Published:** 2023-01-06

**Authors:** Kazunori Nakaoka, Eizaburo Ohno, Naoto Kawabe, Teiji Kuzuya, Kohei Funasaka, Yoshihito Nakagawa, Mitsuo Nagasaka, Takuya Ishikawa, Ayako Watanabe, Takumi Tochio, Ryoji Miyahara, Tomoyuki Shibata, Hiroki Kawashima, Senju Hashimoto, Yoshiki Hirooka

**Affiliations:** 1Department of Gastroenterology and Hepatology, Fujita Health University, Toyoake 470-1192, Aichi, Japan; 2Department of Gastroenterology and Hepatology, Nagoya University Graduate School of Medicine, Nagoya 464-0813, Aichi, Japan; 3Department of Medical Research on Prebiotics and Probiotics, Fujita Health University, Toyoake 470-1101, Aichi, Japan

**Keywords:** early-stage pancreatic ductal adenocarcinoma, transabdominal ultrasonography, endoscopic ultrasound, endoscopic retrograde cholangiopancreatography, computed tomography, magnetic resonance cholangiopancreatography, liquid biopsy

## Abstract

Pancreatic ductal adenocarcinoma (PDAC) can be treated with surgery, chemotherapy, and radiotherapy. Despite medical progress in each field in recent years, it is still insufficient for managing PDAC, and at present, the only curative treatment is surgery. A typical pancreatic cancer is relatively easy to diagnose with imaging. However, it is often not recommended for surgical treatment at the time of diagnosis due to metastatic spread beyond the pancreas. Even if it is operable, it often recurs during postoperative follow-up. In the case of PDAC with a diameter of 10 mm or less, the 5-year survival rate is as good as 80% or more, and the best index for curative treatment is tumor size. The early detection of pancreatic cancer with a diameter of less than 10 mm or carcinoma in situ is critical. Here, we provide an overview of the current status of diagnostic imaging features and genetic tests for the accurate diagnosis of early-stage PDAC.

## 1. Introduction

Pancreatic ductal adenocarcinoma (PDAC) is the fourth-leading cause of cancer-related deaths, and the number of deaths from PDAC will be the second-highest within the next ten years in the United States [1] and it has a dismal prognosis with a 5-year survival rate of less than 10% [2,3]. The American Cancer Society reported that the number of newly diagnosed patients with PDAC was projected to be about 62,210, and the number of deaths was projected to be 49,830 in the United States in 2022 [2]. The mortality of PDAC has been steadily increasing [2]. PDAC is estimated to become the second-leading cause of cancer-related death by 2030 [1,4,5]. The detection of PDAC at an early stage has been challenging. Thus, most cases of PDAC are advanced or metastatic at diagnosis. Efforts have been made to improve the early diagnosis of PDAC. As such, the Japan Pancreatic Society has demonstrated the diagnostic algorithm of PDAC (Figure 1) and the risk factors for the disease [4] as a strategic approach to screening high-risk patients of the disease [6]. This review discusses the early diagnosis of PDAC, focusing on conventional approaches: transabdominal ultrasonography (TUS), endoscopic ultrasound (EUS), endoscopic retrograde cholangiopancreatography (ERCP), computed tomography (CT), and magnetic resonance imaging/magnetic resonance cholangiopancreatography (MRI/MRCP). Furthermore, we discuss the potential use of liquid biopsies for the early diagnosis of PDAC.

### 1.1. The Importance of Diagnosing Early-Stage Pancreatic Ductal Adenocarcinoma

The 5-year survival rates of patients with PDAC were as follows: PDAC in situ, 85.8%; tumor diameter ≤ 10 mm (Tumor size 1a(TS1a)), 80.4%; and tumor diameter ≤ 20 mm with no lymph node metastasis, 68.7%, with a relatively good prognosis [7]. Since the prognosis of PDAC is not always better than that of other neoplastic diseases [2], the diagnosis of PDAC at an early stage is critical for curative treatments and improving prognosis. In particular, it is crucial to diagnose PDAC when its diameter is still 10 mm or less (≤TS1a). However, Union for International Cancer Control (UICC) stage 0 and IA patients account for only 1.7% and 4.1% of the total PDAC patients [7]. Moreover, 75% of these patients do not exhibit symptoms at the time of diagnosis [8]. These results indicate that the early diagnosis of PDAC has been facing challenges.

### 1.2. The Characteristics of Imaging and Genetic Tests for Early-Stage Pancreatic Ductal Adenocarcinoma

The PDAC patients diagnosed early in stages 0 and 1 are increasing in Japan (Table 1) [9]. Kanno et al. reported the features of early-stage PDAC in Japan [9]. In this report, of the 51 cases of stage 0 PDAC, the frequency rates of tumor identification according to the technique used were as follows: 8.8% for the TUS, 10.0% for the CT, 10.0% for the MRI, and 24.4% for the EUS. These results indicated that, in most cases, identifying a mass is difficult. On the other hand, the frequency rates of the identification of pancreatic duct dilation were as follows: 76.5% for the TUS, 72.0% for the CT, 73.9% for the MRI, and 85.4% for the EUS. These results suggested that pancreatic duct dilation was often observed at the stage of carcinoma in situ, even if the small PDAC cannot be visualized. Furthermore, the frequency rates of tumor identification in 149 stage 1 PDAC cases were as follows: 67.3% for the TUS, 65.8% for the CT, 57.5% for the MRI, and 92.4% for the EUS. In stage 1 PDAC, which progresses to invasive cancer, a tumor mass could be easier to identify compared to stage 0 PDAC, and the frequency rates of the identification of main pancreatic duct dilation were as high as 74.3% for the TUS, 82.2% for the CT, 85.8% for the MRI, and 89.4% for the EUS. The identification of pancreatic duct dilation is critical to the early detection of pancreatic cancer via an imaging examination in most cases. It has been recently reported that the dilation and stenosis of the main pancreatic duct, pancreatic cysts, and local fat changes in the pancreas indicated by the imaging findings are characteristic of early-stage PDAC [10,11,12]. Early diagnosis is essential to improving the prognosis of patients with PDAC; however, this task is considered difficult and thus remains a challenge. Hruban et al. first reported a genetic progression model from the precursor lesions named pancreatic intraepithelial neoplasia (PanIN) to PDAC, and according to their progressions, the extent of atypia was classified as low- and high-grade dysplasias [13]. Cases of PDAC in situ were classified as high-grade dysplasia. Somatic alterations are observed: KRAS mutation and CDKN2A abnormalities are observed in low-grade PanIN, and TP53 and SMAD4 abnormalities are observed in high-grade PanIN [14]. Invasive cancer develops at least a decade after the initiation of mutation in PanIN [15], indicating a large window of opportunity for diagnosis at stage 0 or stage I PDAC and clinical intervention.

## 2. The Opportunities for the Diagnosis of Early-Stage Pancreatic Ductal Adenocarcinoma in Diagnostic Imaging Examination

### 2.1. The Characteristics and Ingenuity of TUS for Early-Stage Pancreatic Ductal Adenocarcinoma

In Japan, most early-stage PDACs were accidentally identified via abdominal ultrasonography at medical checkups or while screening for other diseases [9,10]. TUS is less invasive and widely used; however, the degree of pancreatic imaging in abdominal ultrasonography often depends on the patient’s body shape and gastrointestinal gas status. Furthermore, this technique often makes it challenging to observe the entire pancreas. Patients should be placed in a half-sitting position to observe the entire pancreas via abdominal ultrasonography. This position allows the liver to hang down and be located in front of the pancreas, acting as an acoustic window, thus reducing the influence of intestinal gas. Probe compression may also be applied to patients with a large amount of subcutaneous fat. Additionally, because the pancreas has a relatively large range of motion, postural changes can be actively added. In particular, the visual observation of the pancreas can be improved by taking the right lateral decubitus position for the tail of the pancreas and the left lateral decubitus position for the pancreatic hook. However, the tail of the pancreas is often difficult to visualize because of gastric gas. Thus, drinking 300–700 mL of degassed water would be beneficial (Figure 2).

### 2.2. The Characteristics of CT and Magnetic Resonance Cholangiopancreatography for Early-Stage Pancreatic Ductal Adenocarcinoma

The imaging tests for PDAC are generally centered on CT and/or MRI/MRCP [16]. Dynamic CT is crucial for CT examination. In our facility, the pancreatic parenchymal, portal venous, and venous phases are imaged 45, 70, and 180 s after the administration of a high-concentration contrast medium, and constituent images, such as oblique coronary cluster images, are created in each layer. On the other hand, there is no standardized protocol for MRI examination, unlike contrast-enhanced MRI for the liver, but it is desirable to take images using a 3Tesla imaging device. MRI tests include T1-weighted (T1WI), T2-weighted (T2WI), diffusion-weighted (DWI), and MRCP scans. PDAC exhibits prominent fibrosis inside the tumor and invades around it. Thus, it is visualized as a hypovascular tumor in the parenchymal and venous phase, and the boundary is often unclear via dynamic CT. The MRI test results indicated a low signal on T1WI, a faint low to high signal on T2WI, a high signal on DWI, and a decreased apparent diffusion coefficient (ADC). In the MRCP examination, pancreatic duct disruption and upstream dilation were observed (Figure 3a). Most cases of small PDAC are asymptomatic, so the PDAC is characterized by the presence of pancreatic cysts and the dilation of the main or branched pancreatic ducts accidentally detected via TUS or CT. It has been reported that small PDAC, even in PanIN, has pancreatic parenchymal atrophy and fat replacement around tumors (Figure 3b). These findings have attracted attention, as they may allow for the early detection of PDAC [9,17]. It is speculated that these findings may be concomitant changes associated with stenosis or the occlusion of the branched pancreatic duct associated with PanIN lesions or small PDAC or changes associated with cancer-induced fibrosis. As shown above, in the CT/MRI imaging diagnosis of microscopic PDAC, the important findings in the image diagnosis of small PDAC are the changes in the main pancreatic duct, the atrophy of the pancreatic parenchyma/fat infiltration, and/or the appearance of small cysts. However, in either case, PDAC cannot be diagnosed via imaging examinations alone. If pancreatic duct dilation is observed, further pathological examination is required for the site of obstruction or stenosis as well as for localized pancreatic atrophy or fat infiltration. However, the pathological diagnosis of a small pancreatic tumor is not always possible, and there are a certain number of cases in which a pancreatic tumor is suspected by imaging diagnosis such as CT and MRI, but a definitive pathological diagnosis cannot be made, and a definitive diagnosis is made only after a pancreatectomy operation. As the limitation of the image examinations, localized atrophy and fat infiltration of the pancreatic parenchyma are observed in a certain number in daily examinations with CT or MRI, including mild atrophy and fat infiltration, and there is no coherent report such as the frequency of the findings leading to the diagnosis of PDAC. The problem is that there are no diagnostic criteria for pancreatic atrophy and fat infiltration. Recently, Maxim et al. has reported on the use of imaging techniques such as MRI to evaluate the status of fatty pancreatic disease (FPD), characterized by excessive intra-pancreatic fat deposition (IPFD), suggesting a link to the pathogenesis of type 2 diabetes [18]. IPFD is a pathological finding associated not only with diabetes but also with pancreatitis and early-stage pancreatic cancer. The future analysis of many such reports may establish a new diagnostic method using image analysis technology and lead to the development of technology for the early detection of pancreatic cancer.

### 2.3. The Characteristics of Endoscopic Retrograde Cholangiopancreatography for Early-Stage Pancreatic Ductal Adenocarcinoma

The detection of pancreatic duct changes with ERCP and further tests, such as pancreatic juice cytology, have already been established for the early diagnosis of PDAC [19]. With ERCP, pancreatography reveals a typical pancreatic duct image of invasive PDAC characterized by pancreatic duct disruption. On the other hand, in the case of PDAC in situ, the pancreatic duct image is narrowed but slightly patent and is often accompanied by dilation toward the caudal side. ERCP has been used in diagnosing pancreatic lesions; however, in recent years, due to the concern that post-ERCP pancreatitis can sometimes be a severe contingency and particularly fatal, ERCP tests for the purpose of the diagnostic imaging of pancreatic cancer have declined. Today, instead of ERCP testing, diagnostic tests such as MRCP and EUS are frequently performed for the pancreatic duct. When a histopathological diagnosis of pancreatic cancer is required, EUS-guided fine needle aspiration (EUS-FNA) is performed subsequently, as opposed to pancreatic juice cytology with ERCP, owing to its high diagnostic accuracy for PDAC patients [8]. However, diagnosing PDAC without a mass-forming lesion has been challenging using various imaging modalities. Because stage 0 PDAC, referred to as carcinoma in situ, is a pre-stage lesion, it is not surprising that no tumor mass is detected. Even if a mass is observed, inflammation and fibrosis that occurred around carcinoma in situ may be detected. Thus, it may be challenging to make a histopathological diagnosis with EUS-FNA. In the last decades, certain PDAC cases have been diagnosed via pancreatic juice cytology [20]. Iiboshi et al. first reported the high accuracy of diagnosing PDAC in situ via repeated cytology using pancreatic juice obtained from the endoscopic nasopancreatic drainage (ENPD) [21]. When pancreatic duct stenosis, caliber change, and pancreatic branch duct dilation are observed, serial pancreatic-juice aspiration cytologic examination (SPACE) is recommended for histopathological diagnosis [6]. SPACE is essential for diagnosing early-stage PDAC without a mass-forming lesion using various image modalities (Figure 4). Furthermore, it has been reported that atypical pancreatic duct epithelium can be confirmed by probe-based confocal laser endomicroscopy (pCLE) with ERCP, which is useful for diagnosing pancreatic and early pancreatic cancer [22]. In recent years, genetic tests in which specimens are collected via pancreatography and brush cytology have gained considerable attention. Yokode et al. reported that KRAS mutation was detected in nine of ten cases of pancreatic juice cytology with high-grade PanIN using ENBD, with a low incidence of p53 overexpression and a loss of SMAD4 [23]. Hosoda et al. also suggested the inactivation of Tp53 or SMAD4 with high-grade PanIN [24]. Okada et al. reported that a KRAS mutation was detected in the pancreatic juice of a patient without evidence of malignancy using digital PCR [25]. These findings indicate that molecular testing combined with SPACE should be established to diagnose early-stage PDAC.

### 2.4. The Characteristics of EUS for Early-Stage Pancreatic Ductal Adenocarcinoma

EUS is a detailed test conducted when some tests on other images detect an abnormality in the pancreas. The EUS test can detect PDAC without changes in the main pancreatic duct, and when a tumor mass is observed in the pancreas, a histological diagnosis can be made with the consent of EUS-FNA. Kitano et al. have reported that the EUS test can detect small masses with a sensitivity of over 80%, which is higher than those in the other imaging methods: TUS (17–70%), CT (33–75%), and PET (50%), suggesting that the EUS test has a high sensitivity for the detection of solid pancreatic masses. Contrarily, distinguishing pancreatic cancer from other diseases is difficult on EUS imaging alone [8]. Izumi et al. reported that EUS imaging revealed a hypoechoic area around the main lesion in 9 (56%) of 16 cases of carcinoma in situ, and, histologically, inflammation around the main pancreatic duct of the main lesion was observed in all 16 cases [26]. This finding indicates that inflammation and fibrosis may occur in the pancreatic parenchyma around the main lesion, which may be recognized as a hypoechoic area. Furthermore, the contrast-enhanced harmonic EUS test has a sensitivity of 91.2% and specificity of 94.4%, which are comparable to those of CT for diagnosing PDAC with a diameter of 2 cm or less (sensitivity of 70.6% and specificity of 91.9%) [27]. In general, detecting PDAC with a diameter of 10 mm or less is challenging, but EUS is considered useful for such microscopic PDAC (Figure 5). EUS-FNA is considered the most effective test for the definitive diagnosis of PDAC, as it can diagnose histopathology. The sensitivity and specificity rates of EUS-FNA for the diagnosis of pancreatic cancer were 85–92% and 96–98%, respectively, in four meta-analyses [8]. Even if the tumor diameter is limited to 10 mm or less, the rate of accurate diagnosis is reported to be 82.5–96.0%, showing that EUS-FNA is an adequate examination even for microscopic PDAC [28,29]. However, in carcinoma in situ, which is difficult to identify, the pathological diagnosis via EUS-FNA applied for preoperative pathologic diagnosis was only 16.7% in patients with stage 0 pancreatic cancer [9]. Thus, EUS-FNA cannot be employed for pancreatic cancers without forming a tumor mass, including carcinoma in situ. Notably, there have been reports on needle tract seeding caused by using EUS-FNA [30]. It remains controversial whether EUS-FNA should be used for microscopic PDAC, which is expected to be completely cured with surgical resection.

### 2.5. Usefulness of EUS Elastography for Early-Stage Pancreatic Ductal Adenocarcinoma

Ultrasound elastography is a technique for imaging and quantifying tissue elasticity. It is categorized into strain elastography (strain-EG), a negative correlation with tissue elasticity, and shear wave elastography (SW-EG), a positive correlation with tissue elasticity [31]. In the field of hepatology, EG has attracted attention as an alternative to liver biopsies, the conventional golden standard diagnostic method for diagnosing liver fibrosis [32]. Giovannini et al. first reported the usefulness of EUS-EG for pancreatic diseases in 2006. They suggested that the sensitivity and specificity of the diagnosis of the localized pancreatic lesion were 100% and 67%, respectively [33]. Consequently, quantitative evaluations, such as the strain ratio and histogram analysis, were attempted, and in the meta-analysis of EUS-EG for PDAC, the sensitivity and specificity rates were 98% (95% CI, 96–99%) and 63% (95% CI, 58–69%) in the qualitative evaluation and 95% (95% CI, 93–97%) and 61% (95% CI, 56–66%) in the quantitative evaluation, respectively. Both results showed high sensitivity [34]. In recent years, a multicenter collaborative study on pancreatic micro mass lesions related to EUS-EG has been reported [35], demonstrating the diagnostic ability of the EUS-EG test for PDAC. The sensitivity, specificity, positive predictive, and negative predictive values were 96% (95% CI, 87–100%), 64% (95% CI, 56–71%), 45% (95% CI, 40–50%), and 98% (95% CI, 93–100%), respectively. This result indicated that the negative predictive value was as high as 98%. In addition, Kataoka et al. reported that EUS-EG can be used for small solid pancreatic lesions (SPLs) to exclude PDAC with a high reliability and concordance for soft-tissue lesions without pancreatic duct dilation [36]. It is recommended that malignant findings be excluded if the lesion is soft in the EUS-EG test (Figure 6). However, there is a significant problem in the reproducibility of EUS-EG, i.e., the value of EUS-EG is completely different depending on the location of the ROI. A shear wave can be measured even in EUS, and the target’s hardness can be shown numerically, which can be an objective index [37,38,39]. These studies are expected to develop in the field of the shear wave elastography of EUS.

## 3. Current Status of the Genetic Tests for the Diagnosis of Early-Stage Pancreatic Ductal Adenocarcinoma

Tumor tissue collected via EUS-FNA or ERCP has been used as the gold standard for diagnosing pancreatic cancer. However, the detection of pancreatic duct stenosis and dilation and solid tumor mass in the pancreatic duct via imaging has been challenging, as previously described.

Currently, liquid biopsy, e.g., circulating tumor DNA (ctDNA), has emerged as a promising prognostic biomarker of PDAC [40,41,42]. For example, ctDNA genotyping detected targetable mutations comparable to tissue genotyping in advanced gastrointestinal cancers, including PDAC [43].

Analyses using liquid biopsy are non-invasive and repeatable compared with ERCP and EUS-FNA. Additionally, liquid biopsy has an advantage over tumor tissue biopsy: the latter often fails to capture tumor heterogeneity, while the former reflects the tumor mutational landscape [42]. Thus, it is expected to be used as a new diagnostic tool. The blood-based liquid biopsy includes tumor tissue-derived and tumor-associated components: circulating tumor cells (CTCs), ctDNA, and extracellular vesicles (EV) containing microRNA and exosomes. We review the potential use of some liquid biopsies for the early detection of PDAC. In this review, for genetic tests, we will mainly focus on CTCs and ctDNA.

### 3.1. Circulating Tumor Cells

CTCs are tumor cells that invade the blood vessels from the primary lesion and circulate in the blood; they are generally considered precursors to metastasis [44]. Because CTCs can directly analyze tumor cells by counting the number of CTCs and characterizing gene mutations [44], it is possible to elucidate the nature of tumors and the mechanism of distant metastasis.

CTC counts have been considered a promising liquid biopsy in solid tumors since the number of CTCs is correlated with a poor prognosis of cancers [45,46]; however, in the case of PDAC, sensitivity for CTCs in PDAC is lower than that of other epithelial tumors, partly due to the large amount of stromal hyperplasia and the small number of epithelial components [47]. For example, the detection rates of CTCs by stage of pancreatic cancer were 0% in stage I, 60.7% in stage II, 78.6% in stage III, and 96.3% in stage IV, indicating the unstable detection of early-stage I/II PDAC patients [48]. Franses et al. reported that circulating epithelial cells were highly detected in patients with intraductal papillary mucinous neoplasms (IPMNs), who are at a high risk for PDAC [49]. Therefore, monitoring circulating epithelial cells in the patients may have a clinical implication for using liquid biopsy as an early diagnostic tool.

### 3.2. Circulating Tumor DNA

Circulating tumor (ct) DNA has gained popularity for cancer diagnostic, prognostic, or therapeutic monitoring applications since its identification in the serum of cancer patients [50]. Non-invasive early-stage pancreatic cancer develops with mutations in KRAS, and these pancreatic cancer precursor lesions are thought to progress to invasive cancer through the inactivation of tumor suppressor genes such as TP53, SMAD4, and CDKN2A4, following the KRAS mutation [51]. Detecting KRAS mutations at an early stage may contribute to the early detection of pancreatic cancer. In this manuscript, we focused on the KRAS gene abnormalities of PDAC in describing ctDNA for the purpose of the early detection of pancreatic cancer.

The KRAS point mutation in ctDNA with PDAC is reported frequently [52], but ctDNA release has been elusive [53,54]. On the other hand, ctDNA is released into circulation during tumor cell proliferation according to a fragment size of 145 bp [55]. As such, ctDNA has advantages in tumor profiling by representing the molecular heterogeneity of tumors compared to biopsy studies.

The clinical use of ctDNA in PDAC has been primarily reported in prognostic relevance [54]. Given that >90% of PDAC cases have mutations at KRAS [56], target deep sequencing and digital PCR are used to detect somatic mutations in plasma ctDNA. For example, Botrus et al. demonstrated a molecular landscape of PDAC using the 73-gene panel and identified therapeutically relevant targets in 48% of patients with PDAC [57]. Serial ctDNA tests using digital PCR showed that KRAS-mutated ctDNA was associated with a poor prognosis in patients with resectable PDAC [58]. These studies suggest that ctDNA detection can be of clinical utility during the disease and the treatment of PDAC.

Expectations for using ctDNA to detect KRAS mutations have been raised in the early diagnosis of pancreatic cancer. KRAS mutation is observed in precancerous pancreatic duct lesions or pancreatic cyst fluids, including PanINs and IPMNs [59,60]. For example, PanIN, a microscopic noninvasive epithelial neoplasm, showed >90% KRAS mutations in the all-grade dysplasia of PanIN, whereas no mutation was observed in normal pancreatic duct samples [59]. Despite the high detection of mutations in biopsies or cite-specific liquid biopsies such as pancreatic cyst fluids, blood-based ctDNA detection remains challenging in the early diagnosis of PDAC. One of the challenges is that the source of ctDNA is tumor cells; thus, early-stage PDAC lacks a source, with less than two variant templates per 342 milliliters of plasma [61]. Additionally, the highly desmoplastic tumor microenvironment of PDAC characterizes the low tumor cellularity in PDAC [62,63]. The short half-life of ctDNA, from several dozen minutes to hours, adds a challenge to the sample collection and analytical sensitivity of ctDNA [64]. Cohen et al. demonstrated that molecular-barcoded amplicon sequencing detected KRAS mutations at 30% (95% CI 24–36%) in patients with resectable PDAC, indicating that, even with sensitive techniques, the detection of mutated KRAS using ctDNA has limited clinical use [61]. A recent attempt to raise the sensitivity and specificity of the early diagnosis of PDAC has been made by combining ctDNA with cancer-associated proteins. Values of carbohydrate antigen (CA) 19-9, a prognostic marker but not for a screening [65], have been integrated with ctDNA or circulating free DNA. For example, CancerSEEK, a pan-cancer screening test, demonstrated 72% sensitivity and >99% specificity in patients with symptomatic PDAC, which combines cancer-associated genes and proteins, including KRAS and CA 19-9 [66,67]. Besides KRAS mutations, runt-related transcription factor 3 (RUNX3) methylation, a potential biomarker in PDAC [68], in the cell-free (cf) DNA of patients with stage I PDAC, showed 77.8% sensitivity and 93.5% specificity by the combination of CA 19-9 [69]. These studies suggest that the combination of blood-based ctDNA and proteins has an advantage for the early diagnosis of PDAC in a complementary manner.

In addition to CTC and ctDNA, tumor-cell-derived extracellular vesicles (tEV) are promising liquid biopsies for the early diagnosis of PDAC [70]. Vesicles in the circulating system have been studied as a diagnostic biomarker for PDAC and as an attractive liquid biopsy because they are more abundant and structurally stable [71]. Melo et al. demonstrated that membrane-anchored glypican-1 (GPC1) in vesicles could distinguish PDAC from benign pancreatic diseases [72]. A recent study by Ferguson et al. described that a single tEV analysis was able to detect KRAS and P53 mutations in 15 of the 16 patients with stage I PDAC; they estimated that, by using a modeling approach, a PDAC size of ~1 cm^3^ is detectable in >90% of patients [73]. The advanced application will probably enhance the early diagnosis of PDAC.

## 4. Summary

Imaging tests such as EUS, ERCP, CT, and MRCP for the early diagnosis of PDAC, including stage 0, have advanced. Since PDAC has developed invasive cancer for several years, MRCP may play an essential role in detecting pancreatic duct abnormalities, and EUS could detect hypoechoic areas with stage 0 PDAC. In the histopathological tests, EUS-FNA and ERCP with the SPACE test may be effective in the definitive diagnosis of stage 0 PDAC. For an increasingly efficient diagnosis of early-stage PDAC, EUS and MRCP can be actively performed in patients at a high risk of PDAC development. Additional EUS-FNA and SPACE also make a pathological diagnosis possible for the accurate detection of PDAC at an early stage. In the future, increased detections on imaging examinations of early pancreatic cancer and the definite diagnosis by histopathology with EUS and/or ERCP will lead to an increased number of diagnoses of early-stage PDAC patients and finally contribute to improving prognosis in patients with PDAC. However, in the case of the histological diagnosis of small tumors, there are limitations in making a definitive diagnosis by taking tumor tissue and making a pathological diagnosis. On the other hand, in order to establish more advanced techniques for the early diagnosis of PDAC, progress in liquid biopsy research will be essential to compensate for the limitations of imaging techniques. Moreover, given that the advancement of liquid biopsies is remarkable, novel applications in liquid biopsies such as CTC, ctDNA, methylated cfDNA, and tEV may contribute to the early diagnosis of PDAC. Although the four major genes including the KRAS, TP53, SMAD4, and CDKN2A4 mutations that occur in PDAC are useful for monitoring the treatment response and course of diagnosed pancreatic cancer, it is difficult to use them for early diagnosis at this stage. In order to diagnose early-stage pancreatic cancer by a blood test, for example, approaches that examine the optimal combination of the four major genes and the discovery of new genes commonly expressed in early-stage pancreatic cancer are awaited. Currently, the following two points are inferred to be key to the research on liquid biopsies. This means an increase in the case reports of molecular diagnosis using biomarkers, mainly KRAS, and the search for new biomarkers that characteristically indicate early-stage PDAC. In addition to imaging and liquid biopsy, it will also be essential to develop digital PCR and other molecular biological analysis techniques as elements other than imaging and liquid biopsy.

In the future, the precise combination of the three elements of diagnostic imaging technology, biomarker diagnosis using liquid biopsy, and molecular biological analysis technology, including digital PCR, will undoubtedly lead to the establishment of novel technologies for the early detection of pancreatic cancer.

## Figures and Tables

**Figure 1 diagnostics-13-00215-f001:**
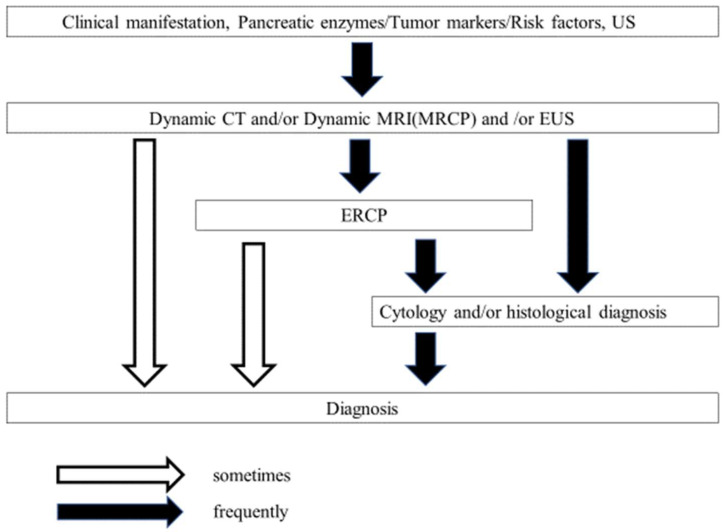
The algorithm for the diagnosis of PDAC (from Reference [6]). US, ultrasonography; CT, computed tomography; ENPD, endoscopic nasopancreatic drainage; ERCP, endoscopic retrograde cholangiopancreatography; EUS, endoscopic ultrasonography; MPD, main pancreatic duct; MRI, magnetic resonance imaging; MRCP, magnetic resonance cholangiopancreatography.

**Figure 2 diagnostics-13-00215-f002:**
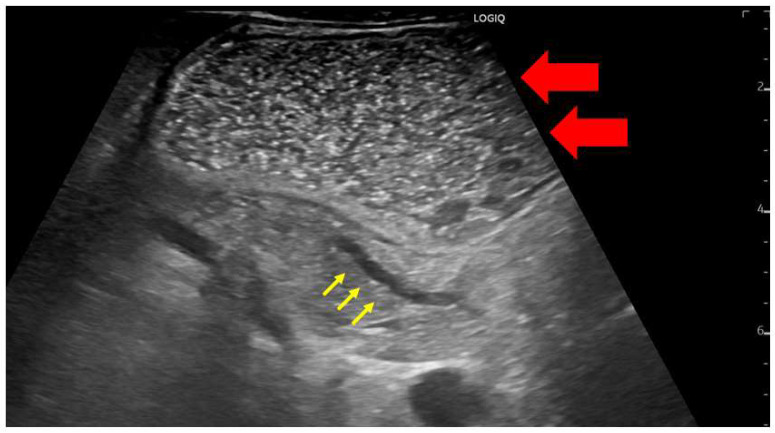
A case in which the main pancreatic duct dilatation was observed by using the drinking water method to fill the gastrointestinal tract with liquid for the observation of the pancreatic body tail. The gastrointestinal tract itself can be used as an acoustic window by instructing the patient to drink degassed water (red arrow); this enables the clear visualization of the tail of the pancreas (yellow arrow).

**Figure 3 diagnostics-13-00215-f003:**
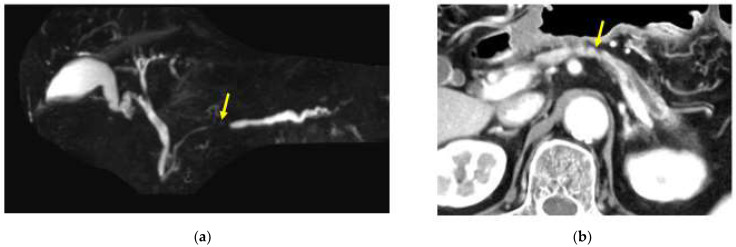
A case in which no pancreatic tumor was found in the pancreatic body by CT or MRI imaging, but atrophy of the pancreatic parenchyma in the caudal part of the pancreatic body and dilation of the main pancreatic duct in the caudal part were observed, which were later diagnosed as in the body. (**a**) Magnetic resonance cholangiopancreatography showing stenosis of the MPD (red yellow arrow) and dilation of the MPD in the caudal part. (**b**) Enhanced CT showing the localized atrophic change (yellow arrow) in the pancreatic body and the dilation of the main pancreatic duct (MPD) in the caudal part.

**Figure 4 diagnostics-13-00215-f004:**
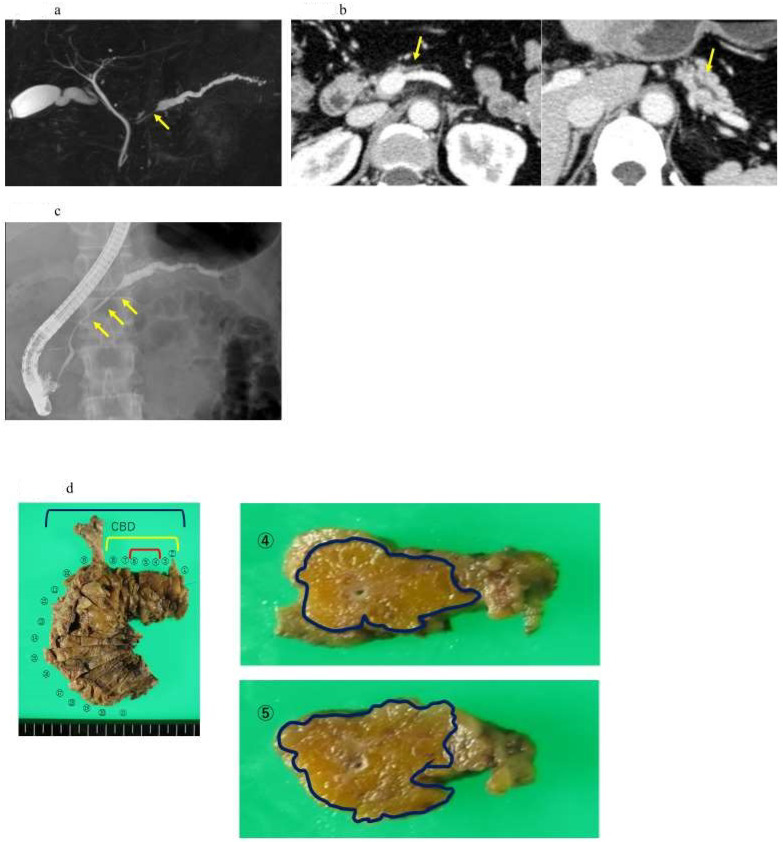
A case of intraepithelial carcinoma of the pancreas, early-stage PDAC, in which no pancreatic tumor was found by CT or MRI imaging, but severe lipidosis of the pancreatic body was observed, leading to the diagnosis of a pancreatic tumor by pancreatic fluid cytology. (**a**) Magnetic resonance cholangiopancreatography showing stenosis of the MPD (yellow arrow) and dilation of the MPD and branch ducts in the caudal part. (**b**) Enhanced CT showing localized atrophic change (yellow arrow) in the pancreatic body and dilation of the MPD in the caudal part. (**c**) Endoscopic retrograde cholangiography demonstrating stenosis of the MPD in the pancreatic body (yellow arrow) and detailed in the caudal part. (**d**) In the resected specimens, severe stenosis of the main pancreatic duct was observed in the resected sections 2–8 (yellow frame), and, in particular, severe steatosis was observed in the resected sections 2–11 (out of the blue frames, 4 and 4). Additionally, 5 shows the resected section. High-grade PanIN was detected in the 4, 5, and 6 resected sections of the pancreatic body (red frame). (**e**) Pathological findings indicated the low papillary proliferation of the pancreatic duct epithelium in the lumen of the main pancreatic duct of pancreatic bodies 4, 5, and 6 and pancreatic lobules associated with chronic pancreatitis in the surrounding pancreatic tissue. Disappearance, fatification, and remaining islets of Langerhans were observed (HE × 40). The pancreatic ductal epithelium exhibited nucleomegaly, a mild chromatin increase, and some clear nucleoli, and high-grade PanIN was detected (HE × 100).

**Figure 5 diagnostics-13-00215-f005:**
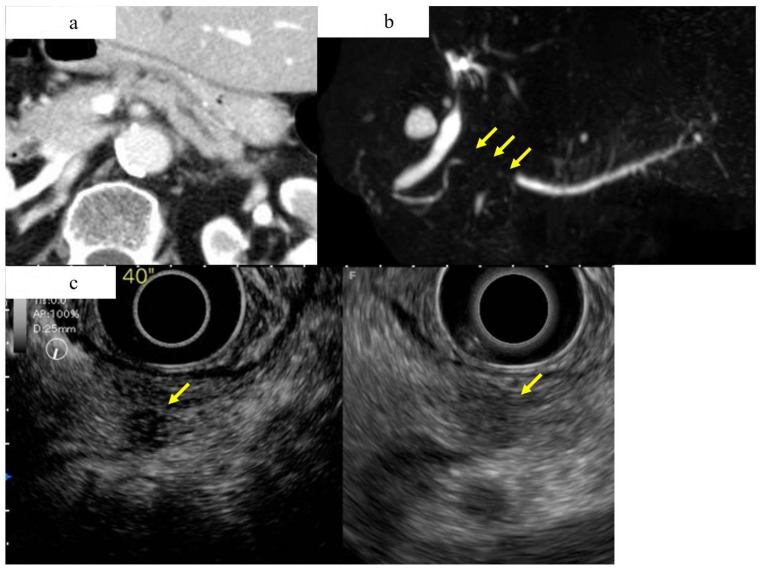
A case of early-stage PDAC in the body in which CT and MRCP imaging studies did not show a pancreatic tumor, but an EUS study showed a pancreatic mass. (**a**) Contrast-enhanced CT showing mild dilation of the main pancreatic duct, but no obvious tumor was detected (yellow arrow). (**b**) Magnetic resonance cholangiopancreatography showing stenosis of the MPD (yellow arrow) and dilation of the MPD and branch ducts in the caudal part. (**c**) A hypoechoic lesion with a diameter of 8 mm (yellow arrow) was detected in the pancreas using fundamental B-mode EUS (**right**), and enhanced-EUS (**left**) revealed that the lesion (yellow arrow) had a lower echo signal intensity than the surrounding pancreatic tissue.

**Figure 6 diagnostics-13-00215-f006:**
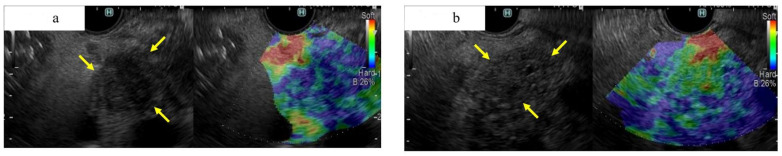
A case of small ductal carcinoma of the pancreatic tail, imaged as a hard mass by EUS elastography (**a**), and a case of autoimmune pancreatitis of the pancreatic tail, imaged as a soft mass by EUS elastography. (**a**) Fundamental B-mode of EUS showing a 13 mm well-defined hypoechoic mass with irregular contours in the tail of the pancreas (yellow arrow). This tumor was shown to be a hard tumor on EUS-EG and was diagnosed as PDAC surgically. (**b**) Fundamental B-mode of EUS showing a 15 mm hypoechoic mass with an indistinct border and irregular outline in the tail of the pancreas (yellow arrow). This tumor was shown to be a relatively soft tumor on EUS-EG and was diagnosed as autoimmune pancreatitis by EUS-FNA.

**Table 1 diagnostics-13-00215-t001:** The imaging findings and modalities for the diagnosis of early-stage pancreatic cancer in patients with PDAC stage 0 and I (from Reference [9]).

Modalities	Findings	All Cases (%) (n = 200)	Stage 0 (%) (n = 51)	Stage I (%) (n = 149)
US		135/200 (67.5)	34/51 (66.7)	101/149 (67.8)
Findings	MPD dilatation	101/135 (74.8)	26/34 (76.5)	75/101 (74.3)
	MPD stenosis	27/135 (20)	2/34 (5.9)	25/101 (24.8
	Tumors	71/135 (52.6)	3/34 (8.8)	68/101 (67.3)
CT		196/200 (98)	50/51 (98)	146/149 (98)
Findings	MPD dilatation	156/196 (79.6)	36/50 (72)	120/146 (82.2)
	Tumors	101/196 (51.5)	5/50 (10)	96/146 (65.8)
	Focal fatty changes	82/196 (41.8)	21/50 (42)	61/146 (41.8)
MRI		173/200 (86.5)	46/51 (90.2)	127/149 (85.2)
Findings	MPD dilatation	143/173 (82.7)	34/46 (73.9)	109/127 (85.8)
	Tumor	78/173 (45.1)	5/46 (10.9)	73/127 (57.5)
EUS		173/200 (86.5)	41/51 (80.4)	132/149 (88.6)
Findings	MPD dilatation	153/173 (88.4)	35/41 (85.4)	118/132 (89.4)
	MPD stenosis	98/173 (56.6)	28/41 (68.3)	70/132 (53)
	Tumor	132/173 (76.3)	10/41 (24.4)	122/132 (92.4)
ERCP		141/200 (70.5)	47/51 (92.2)	94/149 (63.1)
Findings	MPD dilatation	114/141 (80.9)	39/47 (83)	75/94 (79.8)
	MPD stenosis	112/141 (79.4)	39/47 (83)	73/94 (77.7)

US, ultrasound; CT, computed tomography; MRI, magnetic resonance imaging; EUS, endoscopic ultrasound; ERCP, endoscopic retrograde cholangiopancreatography; MPD, main pancreatic duct.

## Data Availability

Data supporting the reported results can be found in the references at the end of this article.

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
