# Peer review of "Current Status of the Diagnosis of Early-Stage Pancreatic Ductal Adenocarcinoma"

_diagnostics, 2023, doi:10.3390/diagnostics13020215_

Round 1

Reviewer 1 Report

Well written review of the current situation.

Author Response

Thank you for your positive feedback on this review. We have made a few additions to the text and summary, including a future outlook.

Reviewer 2 Report

The manuscript by Nakaoka et al. summarizes imaging and genetic-based diagnostic methodologies for the early detection of PDAC tumors (ie, those with a size of 10 mm or less). The topic is interesting and has relevant clinical value in the field, however I found the manuscript in this current version to have little publication relevance.

First of all, the title of the manuscript is a bit of an overstatement: most of the body of the manuscript is about imaging tests but a cursory description of genetics. The structure of the manuscript should be changed to get to the point, in order to have enough space to expand the information on genetic diagnostic tests.

Beyond this, the main problem with this work is that it is merely descriptive. There is little discussion and perspectives.

Author Response

Thank you for pointing this out. As you pointed out, this review did not discuss the diagnosis of early-stage pancreatic cancer enough as it focused on stratified analysis. Hence, we have made the following modifications.

Comments and Suggestions No.1

First of all, the title of the manuscript is a bit of an overstatement: most of the body of the manuscript is about imaging tests but a cursory description of genetics. The structure of the manuscript should be changed to get to the point, in order to have enough space to expand the information on genetic diagnostic tests.

To Answer.1.

Thank you for pointing this out. As you pointed out, this review is mostly concerned with the possibility of detecting early pancreatic cancer by imaging diagnosis. The reason for this is that genetic analysis using liquid biopsy in pancreatic cancer has not yet been fully established. Therefore, in this report, we focus our discussion on KRAS, which may lead to early detection and diagnosis of pancreatic cancer. Since it is expected that liquid biopsy of pancreatic cancer will progress both in terms of research focusing on KRAS and the search for new biomarkers in the future, this report discusses liquid biopsy of pancreatic cancer from the perspective of both researches focusing on KRAS and the search for new biomarkers. Therefore, we have added and modified the summary and discussion as follows in red letters.

(Page 20, Line 5 -13)

In the future, increased detections on imaging examinations of early pancreatic cancer and the definite diagnosis by histopathology with EUS and/or ERCP will lead to an increased number of diagnosis of early-stage PDAC patients and finally contribute to improving prognosis in patients with PDAC. However, in the case of histological diagnosis of small tumors, there are limitations in making a definitive diagnosis by taking tumor tissue and making a pathological diagnosis. On the other hand, in order to establish more advanced techniques for the early diagnosis of pancreatic cancer, progress in liquid biopsy research will be essential to compensate for the limitations of imaging techniques.

(Page 20, Line 15 – Page 21, Line 6)

The following two points are inferred to be key to the research on liquid biopsies. This means an increase in case reports of molecular diagnosis using biomarkers, mainly KRAS, and the search for new biomarkers that characteristically indicate early-stage pancreatic cancer. In addition to imaging and liquid biopsy, it will also be essential to develop digital PCR and other molecular biological analysis techniques as elements other than imaging and liquid biopsy.

In the future, the precise combination of the three elements of diagnostic imaging technology, biomarker diagnosis using liquid biopsy, and molecular biological analysis technology, including digital PCR, will undoubtedly lead to the establishment of novel technologies for the early detection of pancreatic cancer.

Comments and Suggestions No.2

Beyond this, the main problem with this work is that it is merely descriptive. There is little discussion and perspectives.

To Answer.2

Thank you for pointing this out. Adequate discussion was indeed lacking. Therefore, we have added discussion to the following two points as you indicated.

  • For additional discussion on image analysis, we added a discussion on the possibility of early detection of fatty pancreatic disease (FPD) using MRI based on the article on the association between FPD and diabetes mellitus. we have added a manuscript and a reference as follows in red letters.

(Page 9, Line 5 – 11)

Recently, Maxim et al. has reported on the use of imaging techniques such as MRI to evaluate the status of fatty pancreatic disease (FPD), characterized by excessive intra-pancreatic fat deposition (IPFD), suggesting a link to the pathogenesis of type 2 diabetes (18). IPFD is a pathological finding associated not only with diabetes but also with pancreatitis and early-stage pancreatic cancer. Future analysis of many such reports may establish a new diagnostic method using image analysis technology and lead to the development of technology for early detection of pancreatic cancer.

Reference 18

  1. Petrov MS, Taylor R. Intra-pancreatic fat deposition: bringing hidden fat to the fore. Nat Rev Gastroenterol Hepatol. 2022;19(3):153-68.

(ï¼’) The progress of future liquid biopsy research and the potential of novel diagnostic techniques for the diagnosis of early pancreatic cancer have been added to the summary as follows in red letters.

(Page 20, Line 15 – Page 21, Line 6)

The following two points are inferred to be key to the research on liquid biopsies. This means an increase in case reports of molecular diagnosis using biomarkers, mainly KRAS, and the search for new biomarkers that characteristically indicate early-stage pancreatic cancer. In addition to imaging and liquid biopsy, it will also be essential to develop digital PCR and other molecular biological analysis techniques as elements other than imaging and liquid biopsy.

In the future, the precise combination of the three elements of diagnostic imaging technology, biomarker diagnosis using liquid biopsy, and molecular biological analysis technology, including digital PCR, will undoubtedly lead to the establishment of novel technologies for the early detection of pancreatic cancer.

Reviewer 3 Report

I read with great interest the review entitled « current status of the diagnosis of early-stage pancreatic ductal adenocarcinoma » by Kazunori Nakaoka et al. It provides a detailed description of the different approaches with their performance used for the detection of PDAC in early stages. I think that this review is useful tools to better appreciate to challenges of PDAC early detection. However, this review contains minor defaults that I encourage the authors to correct in order to increase the quality of the manuscript (see below). Moreover, the legends of the images should be more detailed for non-specialist readers. A careful reading by an English native speakers is required to rephrase a certain number of sentences. A short description of the approaches at the beginning of each paragraph (for example like in paragraph “Usefulness of EUS elastography….”) would greatly improve the understanding for a non-specialist reader. To be complete, I would have added at the end of the review a small paragraph describing the high risk populations of which these early detection methods could be applied.

Minor comments :

Page 2 line 52: acronym of TS1a.

Page 2 line 54: a space before « the diagnosing » is missing.

Page 2 line 54: diagnosis instead of diagnosing.

Figure 1: the acronym of US is not mentioned.

Table 1: the different risk factors in Table 1 could be accompanied by a major reference.

Page 3 line 57: acronym of UICC.

Figure 2: add arrows on the images to indicate the main features and organs. If you are not a specialist, you don’t know what to look.

Page 5 line 121: “on their hand “ ?

Page 5 line 125: “grows”, please verify.

Page 5 line 132: dot after “incidentally”.

Page 5 line 144: tumors (plural)

Figure 3: same comments as Figure 2.

Page 6 line 155: the last sentence is not clear, please rephrase it.

Page 6 line 161: has instead of have (the detection is the subject)

Page 6 line 170-173: the sentence is not clear, please rephrase it.

Page 6 line 182: acronym of SPACE.

Figure 4: same comments as Figure 2.

Page 9 line 238-240: Figure 4: same comments as Figure 2.

Figure 5: same comments as Figure 2.

Page 10 line 267: SPL acronym.

Page 10 line 29: CTCs instead of CTCS.

Page 10 line 302-303: the sentence is not clear, please rephrase it.

Page 10 line 306: the sentence is not clear, please rephrase it.

Page 11 line 318-321: the paragraph is not clear, please rephrase it.

Page 11 line 330: tests instead of testes.

Page 11 line 342: the sentence is not clear, please rephrase it.

Author Response

Response to Reviewer 3 Comments

Reviewer 3

Comments and Suggestions No.1

This review contains minor defaults that I encourage the authors to correct in order to increase the quality of the manuscript (see below).

To Answer.1

Thank you for pointing this out. We have corrected the minor comment as follows.

Page 2 line 52: acronym of TS1a.

We corrected that “Tumor size 1a (TS1a)”(Page 4 line 3).

Page 2 line 54: a space before « the diagnosing » is missing. ã€€

We modified (Page 4 line 5).

Page 2 line 54: diagnosis instead of diagnosing.

We corrected (Page 4 line 5).

Figure 1: the acronym of US is not mentioned.

We added the acronym US, ultrasonography in Figure 1.

Table 1: the different risk factors in Table 1 could be accompanied by a major reference.

Thank you for your comment. We removed Table 1 for the different risk factors.

Page 3 line 57: acronym of UICC. 

We corrected that “Union for International Cancer Control (UICC)” (Page 4 line 8).

Figure 2: add arrows on the images to indicate the main features and organs. If you are not a specialist, you don’t know what to look.

Thank you for your comment, we added the part of pancreatic duct dilatation and fluid in the gastrointestinal tract are indicated by arrows.

Page 5 line 121: “on their hand “? 

Thank you for pointing this out. We have corrected it.

Page 5 line 125: “grows”, please verify. 

We modified it to invades.

Page 5 line 132: dot after “incidentally”. 

We added “Dot”

Page 5 line 144: tumors (plural)

We modified it to tumors.

Figure 3: same comments as Figure 2. 

Thank you for your response. In response to your suggestion, we have revised the image legend as follows, adding a description of each case and annotations with arrows so that non-specialist readers can understand it.

Page 34 line 6-13

A case in which no pancreatic tumor was found in the pancreatic body on CT or MRI imaging, but atrophy of the pancreatic parenchyma in the caudal part of the pancreatic body and dilation of the main pancreatic duct in the caudal part were observed, which was later diagnosed as in the body.

(a)       Magnetic resonance cholangiopancreatography showing stenosis of the MPD (red arrow) and dilation of the MPD in the caudal part.

(b)       Enhanced CT showing localized atrophic change (red arrow) in the pancreatic body and dilation of the main pancreatic duct (MPD) in the caudal part.

Page 6 line 155: the last sentence is not clear, please rephrase it. 

We corrected the sentence as follows;

Page 8 line 14-17

There are a certain number of cases in which a pancreatic tumor is suspected by imaging diagnosis such as CT and MRI, but a definitive pathological diagnosis cannot be made, and a definitive diagnosis is made only after a pancreatectomy operation.

Page 6 line 161: has instead of have (the detection is the subject)

We corrected it to “has”.

Page 6 line 170-173: the sentence is not clear, please rephrase it. 

We corrected the sentence as follows;

Page 10 line 4-6

Today, instead of ERCP testing, diagnostic tests such as MRCP and EUS are frequently performed for the pancreatic duct.

Page 6 line 182: acronym of SPACE.

We corrected that “serial pancreatic-juice aspiration cytologic examination (SPACE)” (Page 11 line 2).

Figure 4: same comments as Figure 2.

Thank you for your response. In response to your suggestion, we have revised the image legend as follows, adding a description of each case and annotations with arrows so that non-specialist readers can understand it.

(Page 34 line 16-18)

A case of intraepithelial carcinoma of the pancreas, early-stage PDAC, in which no pancreatic tumor was found on CT or MRI imaging, but severe lipidosis of the pancreatic body was observed, leading to the diagnosis of a pancreatic tumor by pancreatic fluid cytology.

(a) Magnetic resonance cholangiopancreatography showing stenosis of the MPD (red arrow) and dilation of the MPD and branch ducts in the caudal part.

(b) Enhanced CT showing localized atrophic change (red arrow) in the pancreatic body and dilation of the MPD in the caudal part.

(c)Endoscopic retrograde cholangiography demonstrating stenosis of the MPD in the pancreatic body (red arrow) and detailed in the caudal part.

(d)In the resected specimens, severe stenosis of the main pancreatic duct was observed in the resected sections 2–8 (yellow frame), and in particular, severe steatosis was observed in the resected sections 2–11 (out of the blue frames, 4 and 4). 5 shows the resected section). High-grade PanIN was detected in the 4, 5, and 6 resected sections of the pancreatic body (red frame).

(e)       Pathological findings indicated low papillary proliferation of the pancreatic duct epithelium in the lumen of the main pancreatic duct of pancreatic bodies 4, 5, and 6 and pancreatic lobules associated with chronic pancreatitis in the surrounding pancreatic tissue. Disappearance, fatification, and remaining islets of Langerhans were observed (HE × 40). The pancreatic ductal epithelium exhibited nucleomegaly, mild chromatin increase, and some clear nucleoli, and high-grade PanIN was detected (HE × 100).

Page 9 line 238-240: Figure 4: same comments as Figure 2.

Thank you for pointing this out. I have detailed it in Figure legend as you pointed out above.

Figure 5: same comments as Figure 2.

Thank you for your response. In response to your suggestion, we have revised the image legend as follows, adding a description of each case and annotations with arrows so that non-specialist readers can understand it.

(Page 36 line 2-11)

A case of early-stage PDAC in the body in which CT and MRCP imaging studies did not show a pancreatic tumor, but an EUS study showed a pancreatic mass.

(a)       Contrast-enhanced CT showing mild dilation of the main pancreatic duct, but no obvious tumor was detected (red arrow).

(b)       Magnetic resonance cholangiopancreatography showing stenosis of the MPD (red arrow) and dilation of the MPD and branch ducts in the caudal part.

(c)       A hypoechoic lesion with a diameter of 8 mm (red arrow) was detected in the pancreas using fundamental B-mode EUS (right) and enhanced-EUS (left) revealed that the lesion (red arrow) had a lower echo signal intensity than the surrounding pancreatic tissue.

Page 10 line 267: SPL acronym. 

We corrected that to “solid pancreatic lesions (SPLs)”(Page 14 line 6).

Page 10 line 29: CTCs instead of CTCS. 

We corrected it to “CTCs”.

Page 10 line 302-303: the sentence is not clear, please rephrase it.

We rephrased it to “Because CTCs can directly analyze tumor cells by counting the number of CTCs and characterizing gene mutations, it is possible to elucidate the nature of tumors and the mechanism of distant metastasis.” 

Page 10 line 306: the sentence is not clear, please rephrase it.

We rephrased it to “CTC counts have been considered a promising liquid biopsy in solid tumors since the number of CTCs is correlated with a poor prognosis of cancers.

Page 11 line 318-321: the paragraph is not clear, please rephrase it.

Thank you for your advice.

(Page 16 line 14- Page 17 line 1)

We corrected as follows: Circulating tumor (ct) DNA has gained popularity for cancer diagnostic, prognostic, or therapeutic monitoring applications since its identification in the serum of cancer patients (50). The KRAS point mutations in ctDNA with PDAC is reported frequently (51), but ctDNA release has been elusive (52, 53). On the other hand, ctDNA is released into circulation during tumor cell proliferation according to a fragment size of 145 bp (54). As such, ctDNA has advantages in tumor profiling by representing the molecular heterogeneity of tumors compared to biopsy studies.

Page 11 line 330: tests instead of testes. 

We corrected it to “tests.”

Page 11 line 342: the sentence is not clear, please rephrase it.

We rephrased it to “thus, early-stage PDAC lacks a source, with less than 2 variant templates per 342 milliliters of plasma.

Comments and Suggestions No.ï¼’

the legends of the images should be more detailed for non-specialist readers.

To Answer.ï¼’

In response to your suggestion, we have modified the legend of each Figure's image as follows: we have added explanatory text to each part and annotated the images with arrows so that non-specialist readers can understand them.

Comments and Suggestions No.3

A short description of the approaches at the beginning of each paragraph (for example like in paragraph “Usefulness of EUS elastography….”) would greatly improve the understanding for a non-specialist reader.

To Answer.3

In response to your suggestion, we have revised the title of the paradigm as follows

(1)

7.Usefulness of EUS elastography for early-stage pancreatic ductal adenocarcinoma(Page 13 line.3)

Comments and Suggestions No.ï¼”

I would have added at the end of the review a small paragraph describing the high-risk populations of which these early detection methods could be applied.

To Answer.ï¼”

 Certainly, not enough information is available regarding the description of early pancreatic cancer using genetic analysis. The reason for this is that genetic analysis using liquid biopsy in pancreatic cancer has not yet been fully established. Therefore, in this report, we focus our discussion on KRAS, which may lead to early detection and diagnosis of pancreatic cancer. Since it is expected that liquid biopsy of pancreatic cancer will progress both in terms of research focusing on KRAS and the search for new biomarkers in the future, this report discusses liquid biopsy of pancreatic cancer from the perspective of both researches focusing on KRAS and the search for new biomarkers. These considerations are added to the summary.

(Page 20, Line 5 -13)

In the future, increased detections on imaging examinations of early pancreatic cancer and the definite diagnosis by histopathology with EUS and/or ERCP will lead to an increased number of diagnosis of early-stage PDAC patients and finally contribute to improving prognosis in patients with PDAC. However, in the case of histological diagnosis of small tumors, there are limitations in making a definitive diagnosis by taking tumor tissue and making a pathological diagnosis. On the other hand, in order to establish more advanced techniques for the early diagnosis of pancreatic cancer, progress in liquid biopsy research will be essential to compensate for the limitations of imaging techniques.

(Page 20, Line 15 – Page 21, Line 6)

The following two points are inferred to be key to the research on liquid biopsies. This means an increase in case reports of molecular diagnosis using biomarkers, mainly KRAS, and the search for new biomarkers that characteristically indicate early-stage pancreatic cancer. In addition to imaging and liquid biopsy, it will also be essential to develop digital PCR and other molecular biological analysis techniques as elements other than imaging and liquid biopsy.

In the future, the precise combination of the three elements of diagnostic imaging technology, biomarker diagnosis using liquid biopsy, and molecular biological analysis technology, including digital PCR, will undoubtedly lead to the establishment of novel technologies for the early detection of pancreatic cancer.

Round 2

Reviewer 2 Report

Dear authors,

I have carefully read this new version of the manuscript that describes the current status of early-stage PDAC diagnosis.

While I greatly appreciate the efforts of the authors in responding to the above comments and suggestions, I regret to communicate that, to me, the current draft of the manuscript does not adequately improve previously highlighted weaknesses.

The work contains many useful elements and is generally interesting, however:

- no changes in the structure of the manuscript were made and the paper remains confusing.

- as KRAS is altered/mutated in a large number of tumors (indeed, chronic pancreatitis as well as other GI tumors harbor KRASmut), the genetic aspects of early PDAC diagnosis cannot only be summarized in the detection of KRAS.

Author Response

Response to Reviewer 2 Comments

To Reviewer 2

Thank you for pointing this out. As you pointed out, the proofreading of this paper was confusing and there was little mention of genetic testing other than KRAS. Hence, we replied to your comment point by point and we have made the revised manuscript.

Comments and Suggestions reviewer.2

No.1

- no changes in the structure of the manuscript were made and the paper remains confusing.

As you pointed out, this paper is still listing imaging examinationss and genetic tests. It is roughly divided into the item of the imaging examinations and the genetic test.

Therefore, we have added the titles as follows in red letters.

(Page 3, Line 95 -96)

  1. The opportunities for diagnosis of early-stage pancreatic ductal adenocarcinoma in diagnostic Imaging examination

(Page 10, Line 305)

  1. Current Status of the Genetic Tests for the Diagnosis of Early-Stage Pancreatic Ductal Adenocarcinoma

No.2

- as KRAS is altered/mutated in a large number of tumors (indeed, chronic pancreatitis as well as other GI tumors harbor KRASmut), the genetic aspects of early PDAC diagnosis cannot only be summarized in the detection of KRAS.

As you pointed out, this paper is mainly referred to KRAS mutations. Although the four major gene mutations that occur in pancreatic cancer are useful for monitoring the treatment response and course of diagnosed pancreatic cancer, it is difficult to use them for early diagnosis at this stage. In order to diagnose early-stage pancreatic cancer by a blood test, for example, approaches that examine the optimal combination of the four major genes and the discovery of new genes commonly expressed in early-stage pancreatic cancer are awaited.

Non-invasive early-stage pancreatic cancer develops with mutations in KRAS, and these pancreatic cancer precursor lesions are thought to progress to invasive cancer through inactivation of tumor suppressor genes such as TP53, SMAD4 and CDKN2A4, following the KRAS mutation. So detecting KRAS mutations at an early stage may contribute to the early detection of pancreatic cancer. In this manuscript, we focused on the KRAS gene abnormalities of PDAC in describing ctDNA for the purpose of early detection of pancreatic cancer.

Therefore, we have added the manuscript and modified the summary as follows in red letters.

(Page 11, Line 343 -348)

non-invasive early-stage pancreatic cancer develops with mutations in KRAS, and these pancreatic cancer precursor lesions are thought to progress to invasive cancer through inactivation of tumor suppressor genes such as TP53, SMAD4 and CDKN2A4, following the KRAS mutation (51). Detecting KRAS mutations at an early stage may contribute to the early detection of pancreatic cancer. In this manuscript, we focused on the KRAS gene abnormalities of PDAC in describing ctDNA.

(Page 12, Line 418 -424)

Although the four major gene including KRAS, TP53, SMAD4, and CDKN2A4 mutations that occur in PDAC are useful for monitoring the treatment response and course of diagnosed pancreatic cancer, it is difficult to use them for early diagnosis at this stage. In order to diagnose early-stage pancreatic cancer by a blood test, for example, approaches that examine the optimal combination of the four major genes and the discovery of new genes commonly expressed in early-stage pancreatic cancer are awaited.

Round 3

Reviewer 2 Report

The  included in the third version of the manuscript do not introduce significant changes in the work, therefore they do not solve the deficiencies indicated in the 2 previous rounds of  revisions.